# Clock-Controlled Mitochondrial Dynamics Correlates with Cyclic Pregnenolone Synthesis

**DOI:** 10.3390/cells9102323

**Published:** 2020-10-19

**Authors:** Melissa Witzig, Amandine Grimm, Karen Schmitt, Imane Lejri, Stephan Frank, Steven A. Brown, Anne Eckert

**Affiliations:** 1Neurobiology Lab for Brain Aging and Mental Health, Molecular & Cognitive Neuroscience, Transfaculty Research Platform, University of Basel, 4002 Basel, Switzerland; melissa.witzig@swisstph.ch (M.W.); amandine.grimm@unibas.ch (A.G.); karen.schmitt@zi-mannheim.de (K.S.); Imane.Lejri@unibas.ch (I.L.); 2Psychiatric University Clinics Basel, Medical Faculty, University of Basel, 4002 Basel, Switzerland; 3Division of Molecular Psychology, Live Sciences Training Facility, University of Basel, 4055 Basel, Switzerland; 4Division of Neuropathology, Institute of Pathology, University Hospital Basel, 4031 Basel, Switzerland; Stephan.Frank@usb.ch; 5Chronobiology and Sleep Research Group, Institute of Pharmacology and Toxicology, University of Zürich, 8057 Zürich, Switzerland; steven.brown@pharma.uzh.ch

**Keywords:** circadian clock, mitochondrial dynamics, neurosteroid, pregnenolone

## Abstract

Neurosteroids are steroids synthetized in the nervous system, with the first step of steroidogenesis taking place within mitochondria with the synthesis of pregnenolone. They exert important brain-specific functions by playing a role in neurotransmission, learning and memory processes, and neuroprotection. Here, we show for the first time that mitochondrial neurosteroidogenesis follows a circadian rhythm and correlates with the rhythmic changes in mitochondrial morphology. We used synchronized human A172 glioma cells, which are steroidogenic cells with a functional core molecular clock, to show that pregnenolone levels and translocator protein (TSPO) are controlled by the clock, probably via circadian regulation of mitochondrial fusion/fission. Key findings were recapitulated in mouse brains. We also showed that genetic or pharmacological abrogation of fusion/fission activity, as well as disturbing the core molecular clock, abolished circadian rhythms of pregnenolone and TSPO. Our findings provide new insights into the crosstalk between mitochondrial function (here, neurosteroidogenesis) and circadian cycles.

## 1. Introduction

Nowadays, it is well known that the circadian clock plays a paramount role in living organisms. It enables adaptation of behavior and physiology, including metabolism, body temperature, sleep–wake cycle and hormone secretion to daily environmental changes like the light/dark cycle [1,2]. The circadian clock shows a hierarchical structure to maintain the rhythm throughout an organism, with the central circadian pacemaker (located in the suprachiasmatic nucleus (SCN)) on top of the hierarchy, controlling the rhythmicity of downstream oscillators in the periphery [2]. The maintenance of a rhythm with a period length of about 24 h is ensured by self-sustained transcriptional–translational feedback loops [3]. This involves a complex molecular machinery resulting in the rhythmic expression of so-called clock genes, including the trans-activating components CLOCK and BMAL1 (brain and muscle ARNT-Like 1), and the trans-inhibiting components period (isoforms PER1-3) and cryptochrome (isoforms CRY1 and 2). 

Neurosteroids, a specific class of steroids characterized by de novo synthesis in the nervous system, were discovered decades ago [4], and are known to exert brain-specific functions [5]. The first step of neurosteroid production takes place in mitochondria with the transfer of cholesterol from the outer to the inner mitochondrial membrane. The translocator protein TSPO is a component of the mitochondrial cholesterol import machinery. Cholesterol is then converted to pregnenolone (P5), the precursor of all steroids [6]. P5 is further metabolized into subsequent neurosteroids either in mitochondria or in the endoplasmic reticulum (ER) major pathways. In the brain, neurosteroids are synthesized by both neurons and glial cells [5]. These molecules play an important role in the nervous system as they can bind either on nuclear receptors to regulate gene expression or act through the activation of membrane-associated signaling pathways to modulate neuronal function [7]. Indeed, neurosteroids—such as pregnenolone, allopregnanolone, or dehydroepiandrosterone—modulate neurotransmission by acting as allosteric modulators of neurotransmitter receptors (e.g., NMDA or GABA receptors) [8]. As they regulate important processes, such as learning and memory, they represent an attractive therapeutic target in brain disorders, including Alzheimer’s disease (AD), epilepsy, and mood disorders [8,9]. Strikingly, the link between neurosteroid synthesis and the circadian clock remains elusive until today. 

By anticipating the time of day, the circadian clock orchestrates numerous processes in the body, especially metabolic reactions, including glucose and fatty acid metabolism [10]. Since mitochondria are known as metabolic hubs, orchestrating a large variety of metabolic processes, it is not surprising that a lot of these processes are under the control of the circadian clock, including mitochondrial dynamics. Mitochondria are remarkably dynamic organelles that fuse and divide in order to maintain a homogenous population of the organelles [11]. On the one hand, mitochondrial fusion allows mitochondria to interact and communicate with each other, and requires the action of two evolutionarily distinct dynamin-related GTPases, mitofusin 1 and 2 (MFN1/2, for fusion of outer membranes), and optic atrophy 1 (OPA1, for fusion of inner membranes). On the other hand, mitochondrial fission allows the renewal, redistribution, and proliferation of organelles involving the dynamin-related protein 1 (DRP1), master fission mediator, as well as fission protein 1 (FIS1). 

We recently provided the first evidence that mitochondrial dynamics are controlled by the clock with a direct control mechanism via phosphorylation of DRP1 at serine 637 [12]. In particular, we showed that mitochondrial energy metabolism (oxidative phosphorylation, adenosine triphosphate (ATP) production) follows a circadian rhythm, and that suppression of DRP1 activity eliminates circadian ATP production. In addition, blocking DRP1 function impaired the core circadian clock, suggesting feedback regulation of mitochondrial metabolism back to the molecular clock.

Mitochondria are not only the powerhouse of cells, but they are also involved in numerous of cellular functions, including steroidogenesis. Hints for a link between steroid synthesis and mitochondrial architecture were shown in peripheral steroidogenic tissues by Duarte and collaborators who demonstrated that mitochondrial fusion directly correlates with increased steroid production, particularly progesterone, in Leydig cells [13,14]. They showed that MFN2 knockdown significantly decreased mitochondrial fusion and steroid production, suggesting that mitochondrial fusion is required for steroidogenesis.

Thus, mitochondrial dynamics seem to play a role in steroid production, while being under control of the circadian clock. However, nothing is known about the implication of the circadian clock in mitochondrial neurosteroidogenesis. Therefore, we decided to first investigate whether mitochondrial neurosteroid production follows a circadian rhythm, and then to explore potential links with the clock-controlled mitochondrial architecture. For this purpose, we used synchronized human A172 glioma cells, which are steroidogenic cells with a functional core molecular clock [15]. Key findings were recapitulated in mouse brain lysates.

## 2. Materials and Methods

### 2.1. Cell Culture

Human A172 glioma cells, A172 glioma cells transfected with luciferase under control of the BMAL1 promoter (generously provided by Steven Brown from the University of Zurich), wild-type Hela cells, and DRP1 KO HeLa cells (generously provided by Stephan Frank from the University of Basel) were cultured in Dulbecco’s modified Eagle’s medium (DMEM) (Sigma, Buchs SG, Switzerland, no. D6429) supplemented with 1% penicillin and streptomycin (Sigma, no. P4333), 1% GlutaMax (Thermo Fisher Scientific, Allschwil, Switzerland, no. 35050061), and 10% foetal bovine serum (FBS) (Sigma, no. F9665) at 37 °C in a humidified atmosphere containing 7.5% CO_2_. Cell passaging was performed twice a week. Human A172 glioma cells transfected with mitochondria-targeted GFP were maintained under the same conditions.

### 2.2. Mouse Brain Homogenates

Drp1^flx/flx CreERT2^ (DRP1 KO) mice were obtained by crossing Drp1^flx/flx^ mice with mice expressing an inducible Cre recombinase transgene under the control of the CamKIIa promoter (Cre+), which is active in the hippocampus and the cortex of adult mice (from the European Mouse Mutant Archive EMMA strain 02125) [16]. At 8 weeks of age, mice were injected intraperitoneally with 1 mg tamoxifen twice daily for five consecutive days to induce recombination of the Drp1 locus [17]. mPer1/mPer2 double-mutant (PER1/PER2) mice were generated as described previously [18]. Mouse brain samples were kindly provided by Jürgen A. Ripperger and Urs Albrecht (University of Fribourg, Switzerland).

Animal experiments were performed according to standardized experimental designs assessing circadian rhythms in mice [19]. Mice were housed at 24 °C on a conventional 12:12 light/dark cycle until the age of 10–12 weeks. Then, they were placed in constant darkness 5 days before the beginning of the experiment with free access to ordinary food (normal chow) and water. Indeed, organisms kept under constant condition (12:12 dark/dark condition) display persisting and intrinsic circadian rhythms (of about 24 h) that may differ from daily rhythms (rhythms synchronized by external clues, such as light/dark cycles). Under constant dark conditions, the time is expressed in circadian time (CT) units, where CT0 is the start of subjective daytime and CT12 is the start of subjective night-time (corresponding to the onset of activity for nocturnal animals like mice) [19]. Animals were sacrificed every 4 h over 24 h and brains were harvested. All experiments were performed in accordance with Swiss animal protection legislation and with approval of the Basel Veterinary Committee for Animal Care: permits 2393 and 23288. The date when the ethical approval was obtained is 22 February 2012 for both permits.

Brain homogenates of wild-type mice, PER1/PER2 and DRP1 knock-out mice were prepared as previously described [12]. Brains were dissected on ice and washed in ice-cold buffer (210 mM mannitol, 70 mM sucrose, 10 mM Hepes, 1 mM EDTA, 0.45% BSA, 0.5 mM DTT, and Complete Protease Inhibitor mixture tablets (Roche Diagnostics, Sigma, no. 11836153001)). After removing the cerebellum, tissue samples were homogenized in 2 mL of buffer with a glass homogenizer (10–15 strokes, 400 rpm), and protein concentration was determined before pregnenolone and western blot analysis.

### 2.3. Transfection with Mitochondria-Targeted GFP

To visualize mitochondrial structures, A172 glioma cells were transfected with plasmid DNA of a mitochondria-targeted GFP (mito-GFP, mitochondria targeting sequence in pEGFP-N1 plasmid for mammalian expression) [20] using transfection reagent Lipofectamine 2000 (Thermo Fisher Scientific, no. 11668027). Cells were seeded 24 h before transfection at a density of 5 × 10^5^ cells in 24-wells plates (Falcon, Amsterdam, The Netherlands, no. 353047) to reach a confluency of 70–90%. At the day of transfection, medium was exchanged with OptiMEM I reduced serum medium (Thermo Fisher Scientific, no. 31985070) containing 5% FBS and no antibiotics. Plasmid DNA was diluted in OptiMEM to a final concentration of 5 μg. Then, OptiMEM containing 4 μL Lipofectamine 2000 and diluted plasmid DNA were mixed 1:1, incubated for 5 min at room temperature and added to cells. After 6 h incubation at 37 °C, medium was changed to OptiMEM containing 5% FBS. After 48 h medium was changed to selection medium (DMEM with 1% Penicillin Streptomycin and GlutaMax, 10% FBS and 300 μg/mL of antibiotic Geneticin (Thermo Fisher Scientific, no. 11811031). Stably transfected cells were selected for two months.

### 2.4. Circadian Clock Synchronization

To analyze all parameters (mitochondrial morphology, phosphorylation of DRP1 at serine 637, pregnenolone production, and TSPO protein level) in a circadian time dependent manner, circadian clocks of A172 glioma cells were synchronized using a horse serum shock [21]. Cells were incubated 2 h at 37 °C with 50% horse serum in DMEM, then washed three times with PBS and placed in DMEM medium supplemented with 1% penicillin/streptomycin, 1% GlutaMax, and 5% FBS at 37 °C in a humidified atmosphere containing 7.5% CO2. Corresponding experiments were then performed from 8 h to 48 h post-synchronization in 4 h intervals.

### 2.5. MDIVI-1 Treatment Paradigm

For the purpose of this study, the small molecule inhibitor MDIVI-1 (Sigma-Aldrich, Buchs SG, Switzerland, no. MO199) was used to disturb mitochondrial dynamics in two experimental approaches. First, in a non-circadian approach, pregnenolone (P5) was quantified after MDIVI-1 administration. Cells were treated with different concentration of MDIVI-1 and two inhibitors of downstream enzymes of the neurosteroid synthesis pathway to ensure pregnenolone accumulation: abiraterone (MedChem Express, Bromma, Sweden, no. HY-75054) which inhibits 17α-hydroxylase, and trilostane (Sigma-Aldrich, no. SML0141), an inhibitor of 3β-hydroxysteroid dehydrogenase. Treatment was performed in a salt buffer (140 mM NaCl, 5 mM KCl, 1.8 mM CaCl_2_, 1 mM MgSO_4_ × 7H_2_O, 10 mM HEPES, 10 mM D-glucose, and 0.1% BSA, pH adjusted to 7.4) with 1, 5, and 10 μM MDIVI-1, 100 nM abiraterone, and 25 μM trilostane for 2 h at 37 °C. Directly after, pregnenolone concentration was quantified in supernatants (see pregnenolone quantification) and mitochondrial morphology was investigated by confocal microscopy. Secondly, in a circadian approach, MDIVI-1 was applied to chronically disturb mitochondrial network morphology. After synchronization, medium was supplemented with 5 μM MDIVI-1 and after 24 h cells were retreated with 5 μM MDIVI-1 to ensure a chronically disturbed mitochondrial network. During this chronic treatment, cells were harvested from 8 to 48 h after synchronization with 4 h intervals for pregnenolone and TSPO protein quantification. 

### 2.6. Confocal Microscopy

Mitochondrial morphology was assessed in human A172 glioma cells transfected with mito-GFP plasmid. Coverslips were coated with 0.05 mg/mL collagen (BD Biosciences Discovery Labware, Allschwil, Switzerland, no. 354236) for 3 h at room temperature. Coverslips were washed with ultrapure water and cells were added on pre-coated coverslips. Cells were synchronized 24 h later and were fixed 15 min with 1 mL of 2% paraformaldehyde (Sigma-Aldrich, no. P6148) from 8 h to 48 h with 4 h intervals. Cells were mounted on glass slides using mounting medium (Thermo Fisher Scientific, no. P36930) and analyzed with a confocal microscope (Leica DMI4000B confocal microscope, Leica Microsystems (Heerbrugg, Switzerland) with Leica Application Suite Advanced Fluorescence (Leica LAS AF) software version 2.51.6757). Images were acquired using a HCX PL APO objective with a magnification of 63x/1.10-0.60 and oil immersion (Leica Microsystems SPE) with Type F immersion oil (Leica Microsystems, no. 11513859). For mito-GFP detection an excitation beam splitter DD 488/365 was used, and emission was detected in a bandwidth of 508–526 nm. 

### 2.7. Mitochondrial Morphology Quantification

Mitochondrial shape parameters were quantified using the open-source software package ImageJ and as previously described [22]. Briefly, images were background-subtracted (rolling ball radius = 50 pixels) and uneven labeling of mitochondria was improved through local contrast enhancement using contrast-limited adaptive histogram equalization (“CLAHE”). To segment mitochondria, the “Tubeness” filter was applied. After setting an automated threshold, the “Analyze Particles” plugin was used to determine the area and perimeter of individual mitochondria and the “Skeletonize” function was used to measure mitochondrial length.

Three parameters were assessed, namely:-Mitochondrial length: the length reports the mitochondrial length or elongation in pixel, after the mitochondria are reduced to a single-pixel-wide shape (“Skeletonize” function on ImageJ).-Form factor (FF): The form factor value describes the particle’s shape complexity of the mitochondria, as the inverse of the circularity.-Aspect ratio: this parameter is independent of area and perimeter and is defined as the ratio of the major on the minor axis.

### 2.8. Gel Electrophoresis and Immunoblotting

A172 glioma cells were seeded one day before synchronization at a density of 0.15 × 10^6^ cells/mL in DMEM + 1% penicillin/streptomycin + 1% GlutaMax + 10% FBS in 60 mm dishes (Falcon, no. 353004) and were harvested in 100 μL ice-cold protein lysis buffer (150 mM Tris Ultrapure, 150 mM NaCl, 1% Nonidet-P40, 0.1% SDS, 2 mM EDTA) after a washing step with cold PBS^-^ from 8 to 48 h post-synchronization in 4 h intervals. Shortly before harvesting, proteases and phosphatases inhibitors were added to the lysis buffer (complete Mini tablet, Roche Diagnostics, Sigma, no. 11836153001; 1 mM Na_3_VO_4_ and 5 mM NaF). Equal protein amounts of each time points were then used. Cell lysates were added to 12.5 μL NUPAGE LDS sample buffer (4×) (Thermo Fisher Scientific, no. NP0007), 2.5 μL DTT (50 mM) (Sigma-Aldrich, no. 233155) and ultrapure water (final volume 50 μl). Samples were heated for 5 min at 95 °C and centrifuged at 10,000 rpm for 5 min at 4 °C. Gel electrophoresis was performed on the NUPAGE Novex 4–12% Bis-Tris Protein gel, 1.0 mm, 15 well (Thermo Fisher Scientific, no. NP0323BOX) according to the manufacturer’s protocol 

Before performing immunoblotting, membranes were blocked for 1 h at room temperature with blocking buffer consisting of 5% BSA (Sigma-Aldrich, no. A7906) diluted in TBS-0.1% Tween 20 (Sigma-Aldrich, no. 93773). Primary antibodies were used for immunoblotting: anti-PBR (TSPO) antibody EPR5384 (Abcam, Cambridge, UK, no. ab109497), phospho-DRP1 Ser637 (Cell Signaling Technology, Beverly, MA, United States, no. 4867), DRP1 (D6C7) Rabbit mAb (Cell Signaling Technology, no. 8570), and VDAC (Cell Signaling Technology, no. 4866). Membranes were incubated with primary antibody diluted 1:1000 (except for Anti-PBR 1:10000) in antibody incubation buffer consisting of 0.6% BSA in TBS-0.1% Tween 20 and incubated at 4 °C overnight. Then, membranes were incubated with the second antibody against rabbit IgG HRP-linked (Cell Signaling Technology, no. 7074) diluted 1:1000 in antibody incubation buffer for 1 h at room temperature. Protein bands were detected by enhanced chemiluminescent reaction using SuperSignal West Dura Extended Duration Substrate (Thermo Fisher Scientific, no. 34075). For detection, a Gene Gnome Chemiluminescent Imaging System (Syngene) was used. Image analysis was performed using ImageJ (https://imagej.nih.gov/ij/ ImageJ with Java 1.8.0_172).

### 2.9. Pregnenolone Quantification

A172 glioma cells were seeded 24 h before synchronization at a density of 0.15 × 10^6^ cells/mL in DMEM + 1% penicillin/streptomycin + 1% GlutaMax + 10% FBS in 60 mm dishes and were harvested in 250 μL PBS^+^ (Thermo Fisher Scientific, no. 21300-058) and lysed with ultrasound from 8 to 48 h post-synchronization in 4 h intervals. Pregnenolone concentration was quantified in cell lysates and mouse brain lysates with a pregnenolone direct ELISA (DRG, Marburg, Germany no. EIA-4170) according to instructions of the manufacturer. Measurements were then performed with a Cytation 3 Cell Imaging Multi-Mode reader (BioTek, Luzern, Switzerland).

### 2.10. siRNA Transfection

A172 glioma cells were seeded 24 h prior to transfection at 0.15 × 10^6^ cells/mL in 60 mm dishes (Falcon, no. 353004) in DMEM + 1% penicillin/streptomycin + 1% GlutaMax + 10% FBS. Shortly before transfection, medium was changed to OptiMEM I reduced serum medium supplemented with 5% FBS and no antibiotics. For siRNA transfection the transfection reagent Lipofectamine RNAiMax (Invitrogen, Thermo Fisher Scientific, no. 13778150) was used according to the manufacturer’s protocol. First, 12 pmol of each siRNA (see Table 1: siRNA target sequences) was diluted in OptiMEM and gently mixed. Then, 6 μL Lipofectamine RNAiMax transfection reagent was diluted in OptiMEM and incubated for 5 min at room temperature. After incubation, diluted siRNA and RNAiMax were combined 1:1 and incubated for 20 min at room temperature and added to cells (final concentration of 8 nM per siRNA). After 6 h medium was changed to OptiMEM I reduced serum medium supplemented with 5% FBS and 48 h later, cells were synchronized as described and harvested according to the parameter investigated.

### 2.11. Statistical and Circadian Analysis

For statistical analysis Graph Pad Prism Software was used (GraphPad Software Inc. version 5.02, San Diego, CA, USA). Cell culture data are presented as means ± SD, animal data are presented as means ± SEM. One-way ANOVA and Dunnett’s multiple comparison tests were performed to compare all columns to a control group. One-way ANOVA and Tukey’s comparison test was performed to compare all the time points, especially the peak versus trough of a circadian cycle. Student *t*-tests were used to compare the mean of two groups. The statistical differences are represented by the *p*-value: * *p* < 0.05, ** *p* < 0.01, *** *p* <0.001 and **** *p* < 0.0001. 

Data from circadian experiments were considered for further analysis with a specific algorithm for rhythmic transcripts to confirm the circadian nature of the cycles. For this, the JTK-Cycle algorithm implemented in R [23] was used as described [24]. A *p*-value of <0.05 was considered as statistically significant and therefore circadian. When data were considered as circadian, curves were generated in Graph Pad Prism using a standard curve fit function (five parameters).

The complete circadian and statistical analysis, with the corresponding *p*-values, can be find in the Appendix A.

## 3. Results

### 3.1. Mitochondrial Neurosteroidogenesis Shows a Circadian Pattern and is Dependent on a Functional Circadian Clock

First, we investigated P5 production, the precursor of all neurosteroids, in synchronized A172 glioma cells over time. P5 levels showed a 24-h rhythmicity with a significant difference between the minimum, 16 h post-synchronization, and the peak, 28 h post-synchronization **(**Figure 1A, Appendix A). To prove direct control of P5 by the circadian clock, we disturbed the circadian clock using siRNA targeting the clock genes *PER1* and *PER2*, which act as transcriptional repressors in the molecular clock machinery [18]. To this end, we transfected A172 glioma cells with siRNA for *PER1* and *PER2* before cell synchronization. Knock-down efficiency was 80% for *PER1*, and 50% for *PER2* (Appendix A). As additional control, cells expressing luciferase under *BMAL1* promoter control were recorded in the LumiCycle after PER1/PER2 siRNA transfection to assess the functionality of the molecular clock (see also Appendix A). As expected, PER1/PER2 siRNA led to disturbed *BMAL1* expression, reflecting a dysfunctional biological clock (Appendix A). In addition, the 24-h variations of P5 were all abolished in presence of PER1/PER2 siRNA (Figure 1B). Significant changes between former peak (28 h after synchronization) and minimum (16 h after synchronization) were no longer detected (Appendix A).

Next, we performed immunoblotting to determine TSPO protein levels over time (Figure 1C,E). TSPO is a component of the transduceosome involved in cholesterol transfer into mitochondria for the conversion into P5 [6], which constitutes the rate-limiting step of steroidogenesis. TSPO showed significant 24-h variations with a trough 16 h post-synchronization and a peak 28 h post-synchronization in-phase with the P5 rhythm (Figure 1A,C). Of note, the expression of two other components of the transduceosome were also investigated, namely the steroidogenic acute regulatory protein (*StAR*) and cytochrome P450 side-chain cleavage enzyme (*CYP11A1*), but they did not show a significant rhythmic change around the clock (Appendix A). We then checked whether TSPO variations were dependent on a functional clock (Figure 1D,F). Although TSPO still looked rhythmic in the presence of PER1/PER2 siRNA, the statistical analysis revealed that the 24-h variations were abolished (Appendix A). 

To demonstrate the significance of our results in vivo, we measured P5 concentration in brain lysates of wild-type mice kept in constant darkness (Figure 2A). P5 levels exhibited a 24-h rhythm with a peak at CT4 (CT=circadian time) and a trough at CT16. In PER1/PER2 knock-out mice, P5 variations were eliminated when compared to WT mice, as no difference were observed between P5 levels at CT4 and CT16 (Figure 2B, Appendix A). We then performed immunoblotting against TSPO in brain lysates of those mice (Figure 2C). In line with the P5 rhythm, TSPO protein levels exhibited also a 24-h rhythm with a peak at CT4 and a trough at CT16. Again, TSPO variations were eliminated in PER1/PER2 knock-out mice, when compared to WT mice (Figure 2D,F). 

Our in vitro and in vivo results clearly show that P5 and TSPO oscillates in a circadian-like fashion and provide evidence that there is a link between the neurosteroid production pathway and the circadian clock. Furthermore, our findings demonstrate that the 24-h oscillations of P5 and TSPO depend on a functional circadian clock, reflecting direct control by the clock.

### 3.2. Mitochondrial Shape Shows a Circadian Rhythmicity in A172 Glioma Cells

To study whether mitochondria-derived P5 production correlates with mitochondrial network architecture in a circadian context, we assessed mitochondrial morphology over time by visualizing mitochondria in synchronized A172 human glioma cells transfected with mitochondrially-targeted green fluorescence protein (GFP) using confocal microscopy. Mitochondrial network displayed a circadian rhythmicity with a fragmented network 16 h post-synchronization and a tubular network at 28 h post-synchronization (Figure 3A, Appendix A). Accordingly, the mitochondrial length, form factor, and aspect ratio showed a significant difference between the fragmented network 16 h post-synchronization and the tubular network 28 h post-synchronization with less interconnected mitochondria in the fragmented state and interconnected network in the tubular state (Figure 3B, Appendix A).

In line with our previous results [12], we confirmed that phospho-DRP1 (at serine 637), which is the inactive form of fission mediator DRP1, presents a circadian pattern in synchronized A172 glioma cells (Figure 3C,D) with a significant difference between 16 h post-synchronization and the peak 28 h post-synchronization (Appendix A). This correlates with the fragmented and tubular network observed at these time points since low levels of phospho-DRP1 indicate high fission activity, resulting in fragmented mitochondria, whereas high levels of phospho-DRP1 reflect a low fission activity with a highly interconnected network. As already shown in our previous study [12], the 24-h variations of phospho-DRP1 were abolished in presence of PER1/PER2 siRNA (Appendix A), confirming that mitochondrial dynamics are indeed under the control of the molecular clock.

### 3.3. Circadian Changes in Pregnenolone Production Correlate with Rhythmic Mitochondrial Morphology

Based on the data obtained by Duarte and colleagues [14] in Leydig cells (derived from testis, a peripheral endocrine gland), we wanted to assess whether mitochondrial fusion and steroidogenesis were also linked in A172 cells (brain derived cells), and to study whether mitochondria-derived P5 production correlates with mitochondrial network architecture in a circadian context. We used MDIVI-1, a compound known to disturb mitochondrial dynamics leading to a more fused network [25]. We demonstrate that mitochondrial length is increased in a MDIVI-1 dose-dependent manner (Figure 4A), together with an increase of P5 concentration (Figure 4B). Furthermore, the P5 rhythm is in-phase with the circadian change of mitochondrial morphology (Figure 4C). A positive correlation is also observed between P5 levels and mitochondrial elongation (Figure 4D).

Taken together, these results strongly suggest a close link between circadian neurosteroid production and mitochondrial structure.

### 3.4. Disrupted Mitochondrial Dynamics Abolish Rhythmic Variations of Neurosteroidogenesis

To further dissect the link between circadian neurosteroid synthesis and mitochondrial dynamics, we disturbed mitochondrial fusion/fission activity and determined P5 and TSPO levels over time in synchronized A172 glioma cells (Figure 5A–F). To this end, we used two different approaches: a drug-based approach with MDIVI-1 (to inhibit mitochondrial fission) and a molecular approach using MFN2 siRNA (to inhibit mitochondrial fusion) [26]. Of note, MFN2 siRNA reduced *MFN2* expression by about 75% in A172 glioma cells (Appendix A). MDIVI-1 administration as well as MFN2 siRNA both abolished circadian variations of P5 (Figure 5A,B) and TSPO expression (Figure 5C,D). Following MDIVI-1 treatment, P5 (Figure 5A) as well as TSPO (Figure 5C, E) only showed a slight increase over time and no significant changes were detected when compared to the trough (16 h after synchronization) and peak (28 h after synchronization) previously determined in the control condition (Appendix A). Similarly, MFN2 siRNA led to rather flat P5 level (Figure 5B) and a slight decrease in TSPO protein over time (Figure 5D,F). Again, in the presence of MFN2 siRNA, no significant changes between former peak and trough were observed (Appendix A). In addition, we measured P5 concentration in wild-type and DRP1 knock-out (KO) HeLa cells, and showed that, while P5 circadian variations were indeed observed in wild-type cells, they were not present in DRP1 KO cells (Appendix A). Thus, these data, obtained by using different approaches to interfere with mitochondrial dynamics, indicate that circadian variations of neurosteroid production depend on functional mitochondrial dynamics.

To further confirm this finding, we investigated this relationship in vivo comparing P5 concentration and TSPO protein levels in brain lysates of DRP1 KO and wild-type (WT) mice [17] (Figure 5G–I). P5 levels (Figure 5G) and TSPO protein levels (Figure 5H,I) showed a significant difference in WT mice between the peak at CT4 and minimum at CT16, whereas this difference was not detected in DRP1 KO mice (Figure 5G–I).

Hence, these results show that disturbing mitochondrial dynamics abolishes circadian variations in neurosteroid production in vitro and in vivo, indicating that rhythmic changes in P5 production and TSPO levels may result from circadian variations in mitochondrial morphology. Collectively, by linking circadian neurosteroid production to mitochondrial dynamics, our data provide clear evidence that mitochondrial structure and function (here, neurosteroidogenesis) are strongly correlated. Most importantly, we clarified that this link is mediated by the circadian variations in mitochondrial morphology.

## 4. Discussion

We previously showed that mitochondrial dynamics are clock-controlled via circadian regulation of DRP1 phosphorylation and that abrogation of DRP1 activity abolished the rhythmicity of mitochondrial dynamics and the associated mitochondrial bioenergetic functions, such as respiratory activity and ATP production [12]. Here, we aimed to investigate whether mitochondrial neurosteroid production follows a circadian rhythm, and to explore whether clock-controlled mitochondrial dynamics regulate mitochondrial neurosteroid synthesis as well. Our key findings are as follows (see also Figure 6): (i) P5 and TSPO levels exhibit a circadian pattern; (ii) circadian changes in mitochondrial morphology correlate with rhythmic production of P5; and (iii) circadian variations of P5 and TSPO depend on both a functional molecular clock and functional mitochondrial dynamics.

Daily variation in steroid synthesis has already been studied in peripheral tissues. For instance, adrenal synthesis and secretion of glucocorticoids is well-known to follow a circadian rhythm, with a peak around the onset of the active period of the day [27]. Glucocorticoids, such as cortisol, control a variety of physiological processes such as metabolism, cardiovascular activity, immune response, and brain functions. Baburski and colleagues recently showed daily changes in serum testosterone and dihydrotestosterone levels (the two main male sex steroid hormones), as well as in the expression of some steroidogenic related genes—including *StAR*, *CYP11A1*, and *CYP17A1* (Cytochrome P450 17A1 or 17α-hydroxylase)—but not *TSPO*, in Leydig cells of rat testis [28]. Of note, in this study, animals were kept in light/dark conditions (12 h light/12 h darkness), suggesting an influence of light input on the parameters measured. In our study, we showed that *StAR* and *CYP11A1* did not display any significant circadian variations in glioma cells, whereas TSPO levels followed a circadian rhythm in vitro and in vivo (mouse brain lysates). Peripheral tissues are regulated differently by the circadian clock with metabolic information being a key factor influencing the rhythm in peripheral tissues [29] and properties of the circadian molecular machinery, which vary among tissues [30,31]. Hence, components of the steroidogenesis pathway could be regulated in a tissue-specific fashion by the circadian clock explaining the discrepancy in the observations. In addition, there could be subtle differences in regulation of steroid and neurosteroid production. Interestingly, daily variation in neurosteroid levels was first shown by Robel et al. in 1987 who found a persisting rhythm of brain dehydroepiandrosterone (DHEA) and DHEA sulfates (two neurosteroids modulating neurotransmission) levels after removal of steroidogenic peripheral glands (adrenal glands and gonads) [32], confirming the existence of autonomous brain mechanisms coordinating neurosteroid production.

The link between mitochondrial dynamics and steroidogenesis was previously highlighted by Duarte and colleagues [13,14]. They showed that hormone-triggered mitochondrial fusion correlates with increased steroid production, particularly progesterone, in Leydig cells. This study provided first evidence that mitochondrial fusion is an essential step of steroid production, a process which depends on protein kinase A (PKA) activity. Especially, MFN2 expression was upregulated following hormone stimulation, and its knockdown was sufficient to impair steroidogenesis, suggesting an essential role for mitochondrial fusion during steroidogenesis. More recently, Park and colleagues showed that steroid production in Leydig cells correlates with the protein kinase A (PKA)-dependent DRP1 phosphorylation (Ser 637) and mitochondrial elongation [33]. In line with these findings, our study shows for the first time a correlation between steroidogenesis and mitochondrial dynamics in brain-derived cells, in particular a cyclic production of P5, which parallels the circadian rhythm of mitochondrial fusion/fission activity.

In light of our new data, the following mechanisms may explain the circadian rhythm in P5 levels that we observed in our study.

We showed that circadian variations of mitochondrial fusion correlate with rhythmic P5 levels, and that disturbing the molecular clock abolishes both rhythms in mitochondrial dynamics [12] and P5 levels. A link between steroidogenesis and mitochondrial dynamics was previously shown in Leydig cells with mitochondrial shaping proteins, DRP1 and MFN2, playing important roles in the regulation of steroid production [14,33]. Namely, PKA activation by dibutyryl cyclic AMP was shown to induce DRP1 phosphorylation at Serine 637, decreasing DRP1 mitochondrial localization and leading to mitochondrial elongation and steroidogenesis [33]. Strikingly, DRP1 dephosphorylation at Ser 637, which leads to its translocation to mitochondria and mitochondrial fission, is regulated through clock-dependent calcineurin activation [34] with pharmacological inhibition of calcineurin increasing DRP1 phosphorylation and abrogating circadian DRP1 phosphorylation patterns [12]. Based on these findings, we can hypothesize that cyclic steroidogenesis (i.e., P5 circadian synthesis) is regulated by mitochondrial dynamics via clock-controlled calcineurin activity, which governs DRP1 phosphorylation and mitochondrial elongation.

In addition, we showed that when mitochondrial dynamics are impaired (MDIVI-1 treatment or MFN2 siRNA), P5 and TSPO expression become arrhythmic. This suggests that mitochondrial dynamics may modulate TSPO expression. We previously demonstrated that DRP1-mediated mitochondrial bioenergetic oscillations may send signals back to the clock (via adenosine monophosphate-activated protein kinase (AMPK) pathways, sirtuins) thereby modulating parameters of the circadian rhythm, such as period length [12]. Therefore, it is not excluded that this feedback control on the molecular clock influences gene expression, resulting in arrhythmic TSPO expression when mitochondrial dynamics are impaired. Furthermore, when the molecular clock is disturbed (PER1/PER2 knockdown), rhythmic variations of TSPO and P5 are abolished. In line with this observation, TSPO was shown to recruit PKA to mitochondrial to phosphorylate the voltage-dependent anion channel (VDAC1) [35]. One can hypothesize that this mechanism is also involved in the circadian regulation of P5 synthesis, with TSPO recruiting PKA to mitochondria for DRP1 phosphorylation, triggering mitochondrial elongation and steroidogenesis.

Further studies are now necessary to elucidate these underlying mechanisms and to clarify the connection between mitochondrial dynamics, neurosteroidogenesis, and the circadian clock.

Neurosteroids are also able to modulate neuronal bioenergetics by increasing mitochondrial respiration and ATP production, at least in part via nuclear steroid receptor activation [36]. Our recent findings revealed that the P5 levels were decreased in cellular models of AD, together with impairments in mitochondrial bioenergetics [37,38]. Strikingly, treatment with TSPO ligands increased P5 production and improved mitochondrial energy production in the presence of AD-related pathological proteins, amyloid-β peptide, and abnormal Tau protein. These data showed that neurosteroids play an important role in controlling neuronal function by modulating mitochondrial physiology, and that impaired neurosteroidogenesis may be involved in mitochondrial dysfunction underlying neurodegenerative disorders. A better understanding about how neurosteroid synthesis is regulated in the brain may offer the possibility to develop therapeutic strategies against neurodegenerative processes.

## 5. Conclusions

In summary, our new findings provide first insights into the circadian regulation of neurosteroid production. The potential impact of the clock on mitochondrial dynamics is especially important because mitochondrial fusion/fission activity influences crucial mitochondrial functions, including neurosteroid synthesis. Our findings could have multiple implications regarding the regulation of metabolic homeostasis in health and disease states associated with mitochondrial impairments and/or circadian disruption.

## Figures and Tables

**Figure 1 cells-09-02323-f001:**
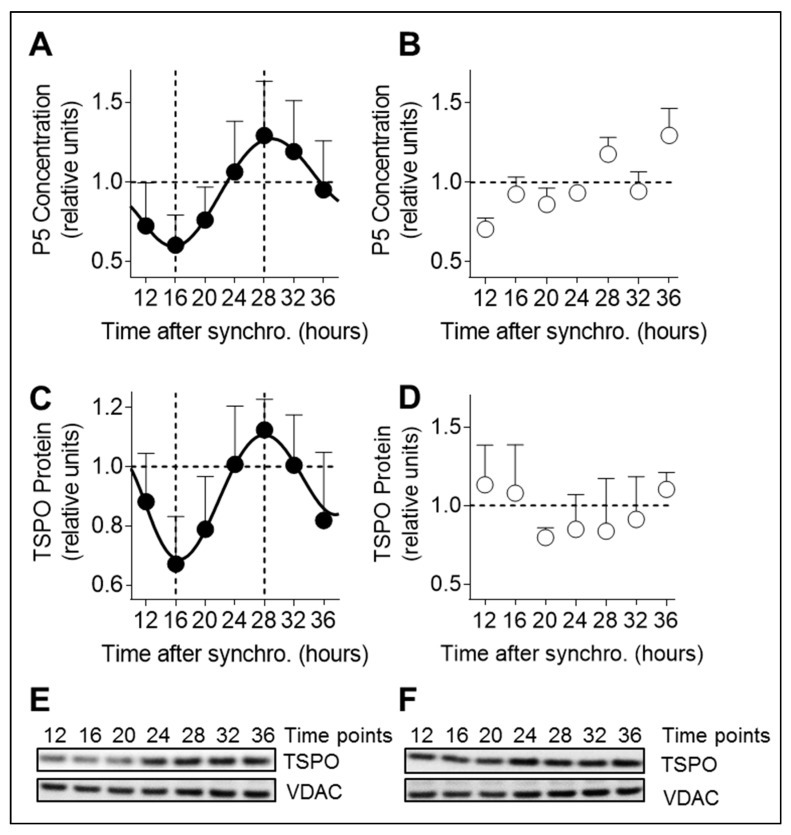
Pregnenolone and TSPO oscillations are dependent on a functional circadian clock in A172 cells. (**A**) P5 (pregnenolone) concentration was quantified in lysates from synchronized A172 glioma cells using a direct pregnenolone ELISA test. Time points from 12 to 36 h post-synchronization are shown. All data are normalized to 1 and presented as mean ± SD (three independent experiments, seven time points, *n* = 12 for each time point). (**B**) P5 concentration in synchronized PER1/PER2 siRNA transfected A172 glioma cells. Time points from 12 to 36 h post-synchronization are shown. All data are normalized to 1 and presented as mean ± SD (three independent experiments, seven time points, *n* = 3–5 for each time point). (**C**) Translocator protein (TSPO) levels were analyzed in cell lysates from synchronized A172 glioma cells by immunoblotting. Time points from 12 to 36 h post-synchronization are shown. All data are normalized to 1 and presented as mean ± SD (three independent experiments, seven time points, *n* = 4–6 for each time point). (**D**) TSPO protein levels in synchronized PER1/PER2 siRNA transfected A172 glioma cells. Time points from 12 to 36 h post-synchronization are shown. All data are normalized to 1 and presented as mean ± SD (two independent experiments, seven time points, *n* = 3–5 for each time point). (**E**) Representative immunoblots of synchronized A172 glioma cell lysates stained with antibodies specific for TSPO and VDAC (voltage-dependent anion-selective channel = control). Time points from 12 to 36 h post-synchronization are shown. (**F**) Representative immunoblots of synchronized PER1/PER2 siRNA transfected A172 cells stained with antibodies specific for TSPO and VDAC (control). Time points from 12 to 36 h post-synchronization are shown.

**Figure 2 cells-09-02323-f002:**
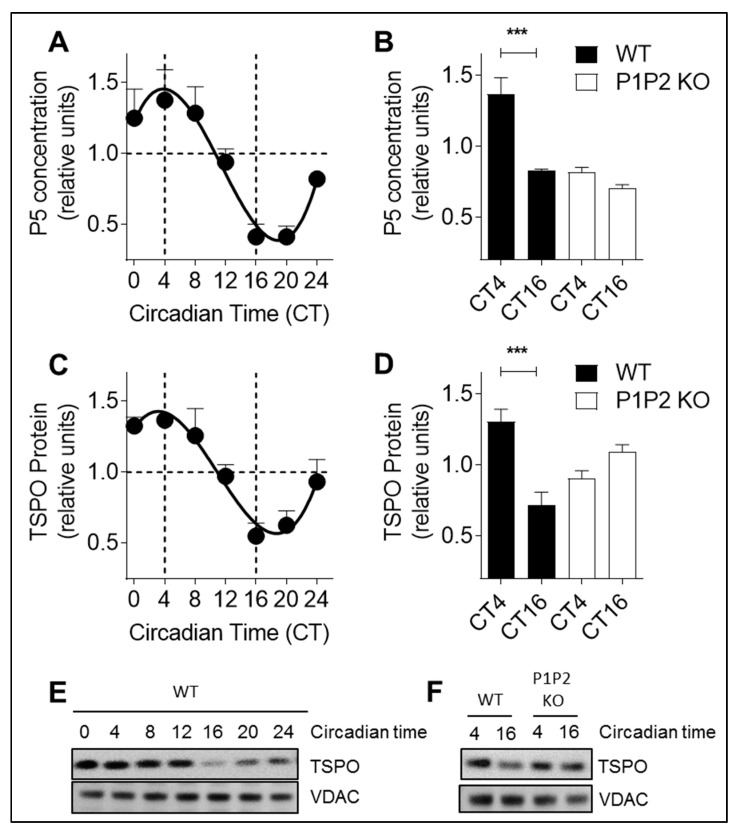
Pregnenolone and TSPO oscillations are dependent on a functional circadian clock in mouse brains. (**A**) P5 concentration was quantified in brain homogenates from non-fasted wild-type mice kept in constant darkness using a direct pregnenolone ELISA test. Circadian times from CT0 to CT24 are shown. All data are normalized to 1 and presented as mean ± SEM (one experiment, seven time points, *n* = 3–4 for each time point). (**B**) Variations in P5 concentration are abolished in PER1/PER2 (P1P2) knock-out (P1P2 KO) mice. P5 levels at peak and trough corresponding to CT4 and CT16 from brain homogenates of wild-type (WT) mice compared to P1P2 KO mice. All data are normalized to 1 and presented as mean ± SEM (WT and P1P2 KO: one experiment, two time points, *n*_WT_ = 8 and *n*_P1P2 KO_ = 3–4 for each time point). *** *p* < 0.001 (unpaired Student’s *t*-test). (**C**) TSPO protein levels were analyzed in brain homogenates from non-fasted wild-type mice kept in constant darkness by immunoblotting. Circadian times from CT0 to CT24 are shown. All data are normalized to 1 and presented as mean ± SEM (one experiment, seven time points, *n* = 2–3 for each time point). (**D**) Variations in TSPO protein levels are abolished in PER1/PER2 (P1P2) knock-out (P1P2 KO) mice. TSPO levels at peak and trough corresponding to CT4 and CT16 from brain homogenates of WT mice compared to P1P2 KO mice. All data are normalized to 1 and presented as mean ± SEM (WT and P1P2 KO: one experiment, two time points, *n*_WT_ = 8 and *n*_P1P2 KO_ = 3–4 for each time point). *** *p* < 0.001 (unpaired Student’s *t*-test). (**E**,**F**) Representative immunoblots of brain lysates from non-fasted wild-type mice kept in constant darkness stained with antibodies specific for TSPO and VDAC (control). (**E**) Circadian times from CT0 to CT24 are shown for wild-type animals (WT). (**F**) Circadian times CT4 and CT 16 (corresponding to peak and trough) are shown for WT and PER1/PER2 knock-out (P1P2 KO) animals.

**Figure 3 cells-09-02323-f003:**
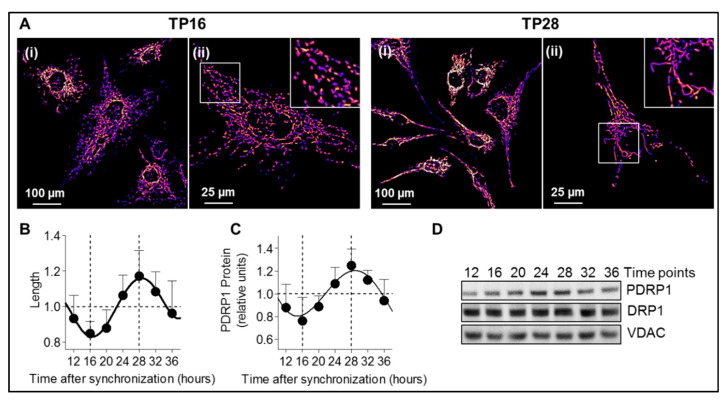
Mitochondrial network morphology shows a circadian rhythmicity in A172 glioma cells. (**A**) Mitochondrial network morphology assessed in synchronized A172 glioma cells transfected with mitochondrially-targeted GFP (false color “fire”) at 16 h and 28 h, corresponding to a fragmented network and a tubular network, respectively. For each representative image, a zoom-in image is shown (400%). Scale bars: 100 µm (i) and 25 µm (ii). The complete cycle (from time point 12 to 36) is shown in Appendix A. (**B**) Quantification of mitochondrial length in synchronized A172 glioma cells transfected with mitochondrially-targeted GFP from 12 to 36 h post-synchronization. All data are normalized to 1 and presented as mean ± SD (two independent experiments, seven time points). About 40 images containing 3000–5000 mitochondria were analyzed per time point. (**C**) Phospho-DRP1 (PDRP1 = dynamin-related protein 1 phosphorylated at serine 637) protein level was analyzed in cell lysates from synchronized A172 glioma cells by immunoblotting. Time points from 12 to 36 h post-synchronization are shown. All data are normalized to 1 and presented as mean ± SD (three independent experiments, seven time points, *n* = 4–6 for each time point). (**D**) Representative immunoblots of synchronized A172 glioma cell lysates stained with antibodies specific for phospho-DRP1 (PDRP1), DRP1 (total DRP1), and VDAC (control). Time points from 12 to 36 h post-synchronization are shown.

**Figure 4 cells-09-02323-f004:**
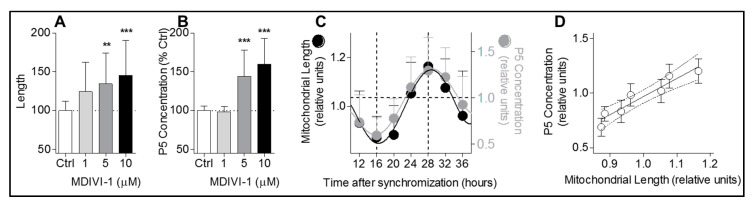
Changes in mitochondrial morphology correlate with rhythmic pregnenolone production. (**A**) Mitochondrial length was quantified after MDIVI-1 (1, 5, and 10 μM) treatment in A172 glioma cells transfected with mitochondrially-targeted green fluorescent protein (GFP). All data are normalized to untreated control (100%) and presented as mean ± SD (two independent experiments, *n* = 33–37 cells for each condition). ** *p* < 0.01, *** *p* < 0.001 (one way-ANOVA with Dunnett’s multiple comparison test versus control (Ctrl)). (**B**) Quantification of P5 concentration in supernatant of A172 cells after MDIVI-1 administration (1, 5, and 10 μM) compared to untreated control. All data are normalized on untreated control (100%) and represent mean ± SD (two independent experiments, *n* = 6–10 for each condition). *** *p* < 0.01 (one way-ANOVA with Dunnett’s multiple comparison test versus control (Ctrl)). (**C**) Mitochondrial length and P5 levels cycles are in-phase. Combined graph of P5 concentration (Figure 1A) and mitochondrial length (Figure 3B). Time points from 12 to 36 h post-synchronization are shown. (**D**) P5 levels correlate with mitochondrial length. Graph represents the values of mitochondrial length around the clock in abscissa versus the values of P5 concentration around the clock in ordinate. Values represent the mean ± SD of each time point. Pearson correlation *r* = 0.9539, *r^2^* = 0.91, *p*-value = 0.0009.

**Figure 5 cells-09-02323-f005:**
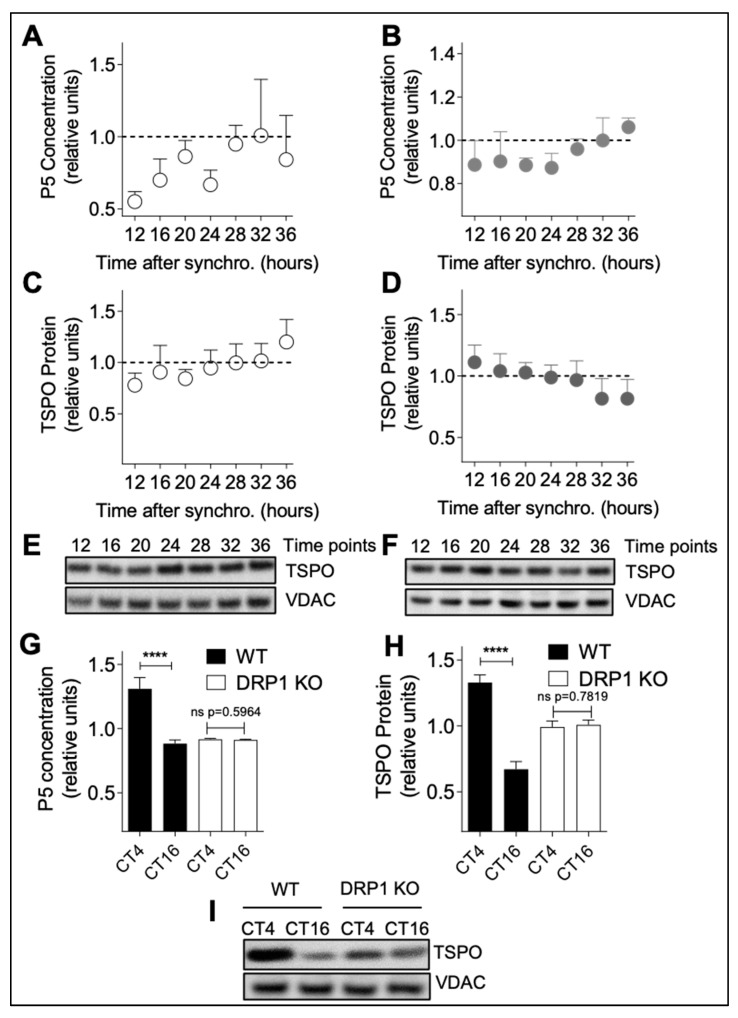
Disrupted mitochondrial dynamics abolish circadian variations of pregnenolone and TSPO. (**A**) P5 concentration was measured in lysates from synchronized A172 glioma cells chronically treated with 5 μM MDIVI-1. Time points from 12 to 36 h post-synchronization are shown. All data are normalized to 1 and presented as mean ± SD (two independent experiments, seven time points, *n* = 4–8 for each time point). (**B**) P5 concentration was measured in lysates from synchronized mitofusion 2 (MFN2) siRNA transfected A172 glioma cells. Time points from 12 to 36 h post-synchronization are shown. All data are normalized to 1 and presented as mean ± SD (two experiment, seven time points, *n* = 4–5 for each time point). (**C**) TSPO protein levels were analyzed by immunoblotting in lysates from synchronized A172 glioma cells chronically treated with 5 μM MDIVI-1. Time points from 12 to 36 h post-synchronization are shown. All data are normalized to 1 and presented as mean ± SD (two independent experiments, seven time points, *n* = 6–8 for each time point). (**D**) TSPO protein expression was analyzed by immunoblotting in lysates from synchronized A172 glioma cells transfected with MFN2 siRNA. Time points from 12 to 36 h post-synchronization are shown. All data are normalized to 1 and presented as mean ± SD (two experiments, seven time points, *n* = 3–6 for each time point). (**E**,**F**) Representative immunoblots of lysates from synchronized MDIVI-1 treated A172 glioma cells (**E**) and MFN2 siRNA transfected cells (**F**) stained with antibodies specific for TSPO and VDAC (control). Time points from 12 to 36 h post-synchronization are shown. (**G**,**H**) Rhythmic changes in P5 levels (**G**) and TSPO protein expression (**H**) are abolished in DRP1 knock-out mice. TSPO levels at peak and trough corresponding to CT4 and CT16 from brain homogenates of wild-type mice compared to DRP1 knock-out (DRP1 KO) mice. All data are normalized to 1 and presented as mean ± SEM (WT and DRP1 KO: two time points, *n*_WT_ = 5 and *n*_DRP1 KO_ = 9 for each time point). **** *p* < 0.0001 (unpaired Student’s *t*-test). (**I**) Representative immunoblots of brain lysates from wild-type (WT) and Drp1 knockout (DRP1 KO) mice stained with antibodies specific for TSPO and VDAC (control). Time points CT4 and CT16 are shown.

**Figure 6 cells-09-02323-f006:**
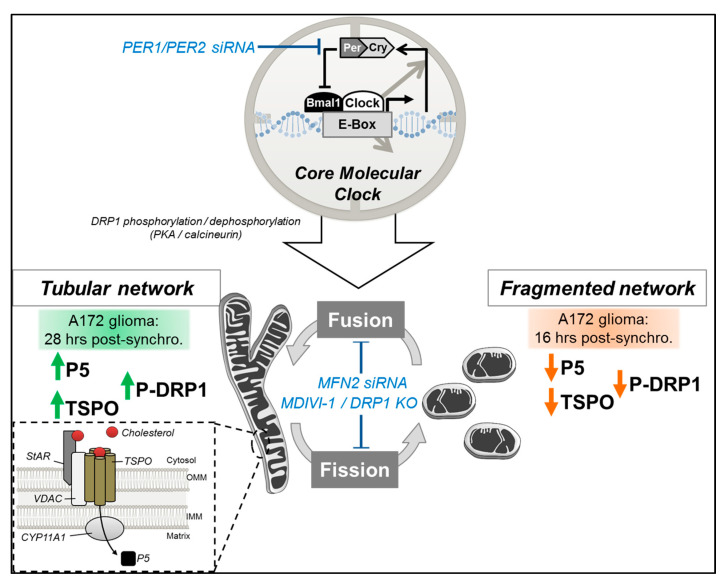
Schematic representation of the relationship between the core molecular clock, mitochondrial dynamics, and neurosteroid synthesis. The clock-controlled mitochondrial morphology correlate with mitochondrial neurosteroidogenesis, with high P5 and TSPO levels detected after fusion (increased P-DRP1 and mitochondrial elongation), and low P5 and TSPO levels after fission (decreased P-DRP1 and mitochondrial fragmentation). Circadian variations of P5 and TSPO are abolished when mitochondrial dynamics are disturbed (e.g., by MFN2 siRNA or MDIVI-1) or when the core molecular clock is affected (siRNA-mediated knockdown of PER1/PER2). CYP11A1: cytochrome P450 side-chain cleavage enzyme, IMM: inner mitochondrial membrane, OMM: outer mitochondrial membrane, MFN2: mitofusin 2, P5: pregnenolone, P-DRP1: dynamin-related protein 1 phosphorylated at serine 637, PER1/PER2: Period 1/2, TSPO: translocator protein, StAR: steroidogenic acute regulatory protein; “T” symbol: inhibition; ↑: increase, ↓: decrease.

**Table 1 cells-09-02323-t001:** siRNA target sequences.

siRNA Name	Number	Target Sequences
Hs_PER1_5	no. SI03076752	CCCAGCGGTTGTCCAGCCCTA
Hs_PER1_4	no. SI00040551	CCCGGACTCTCCACTGTTCAA
Hs_PER1_3	no. SI00040544	CAGCAATGGTTCAAGTGGCAA
Hs_PER1_2	no. SI00040537	CCAGCGCGTCATGATGACCTA
Hs_PER2_11	no. SI03109687	TACGCTGGCCACCTTGAAGTA
Hs_PER2_10	no. SI03075576	CCAGGTTGCATCCATATTTCA
Hs_PER2_9	no. SI03071635	CAGGGTGGGCCCTTTGAATGA
Hs_PER2_6	no. SI02632189	CAGCTGGTCCAGCTTCATCAA
Hs_MFN2_5	no. SI04188835	CTGCACCGCCACATAGAGGAA
Hs_MFN2_8	no. SI04375406	AAGACTATAAGCTGCGAATTA
AllStars Negative Control	no. SI03650318	N/A

PER1: period 1; PER2: period 2; MFN2: mitofusin 2.

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
