# Peer review of "Clock-Controlled Mitochondrial Dynamics Correlates with Cyclic Pregnenolone Synthesis"

_cells, 2020, doi:10.3390/cells9102323_

Round 1
Reviewer 1 Report
In the previous paper, the authors’ group has reported that the morphological change in mitochondria and the level of phosphorylation in mitochondrial protein DRP1 are regulated by the circadian clock. In this paper, they focused on pregnenolone and translocator protein (TSPO) which play a central role in neurosteroidogenesis in mitochondria. The authors revealed that the daily oscillations of pregnenolone and TSPO levels are regulated by the circadian clock. The oscillations are disrupted by the inhibition of the morphological change in mitochondria. These data suggest the link between the circadian clock, mitochondrial dynamics, and steroidogenesis.
The experiments were well- designed with enough controls and the document is easy to understand. Only a few corrections shown below would make this paper better.
Fig.1 and 2: It is not clear how the authors measured the level of P5 and TSPO proteins. Did they quantify these levels by measuring the band's intensities of Western blots?
Fig. 2B and D, Fig. 5G and H: Please show the result of the t-test not only WT but also on P1P2 KO mice (not significant: ns).
P12 L423: Thus, by using different approaches to interfere with mitochondrial dynamics, we show that circadian variations of neurosteroid production depend on functional mitochondrial dynamics.
This seems to be an overstatement at this point since it is possible that MDIVI-1 and MFN2 knockdown could disrupt the daily fluctuation of P5 expression directly without morphological changes of mitochondria.
Author Response
Fig.1 and 2: It is not clear how the authors measured the level of P5 and TSPO proteins. Did they quantify these levels by measuring the band's intensities of Western blots?
Answer: We agree with the reviewer and specified the quantification method in the figure caption:
- Line 294-295: “P5 concentration was quantified in lysates from synchronized A172 glioma cells using a direct pregnenolone ELISA test.”
- Line 300-301: “TSPO protein levels were analysed in cell lysates from synchronized A172 glioma cells by immunoblotting.”
- Line 323-324: “P5 concentration was quantified in brain homogenates from non-fasted wild-type mice kept in constant darkness using a direct pregnenolone ELISA test.”
- Line 330-332: “TSPO protein levels were analysed in brain homogenates from non-fasted wild-type mice kept in constant darkness by immunoblotting.”
Fig. 2B and D, Fig. 5G and H: Please show the result of the t-test not only WT but also on P1P2 KO mice (not significant: ns).
Answer: We thank the reviewer for this comment. We added the advised statistics in the Figure 2B and D, and Figure 5G and H.
P12 L423: Thus, by using different approaches to interfere with mitochondrial dynamics, we show that circadian variations of neurosteroid production depend on functional mitochondrial dynamics.
This seems to be an overstatement at this point since it is possible that MDIVI-1 and MFN2 knockdown could disrupt the daily fluctuation of P5 expression directly without morphological changes of mitochondria.
Answer: We thank the reviewer for making us aware of this point. We showed in Figure 4A that a treatment with MDIVI-1 induces mitochondrial elongation, therefore changing mitochondrial morphology, parallel with an increase in P5 production. In one of our study reference paper (Duarte A. et al, PLoS One 2012), authors showed that MFN2 siRNA decreased mitochondrial fusion as well as the ability of MA-10 cells to synthetize steroids. Thus, it is more likely that circadian variations of P5 depends on circadian variations of mitochondrial dynamics. Nevertheless, we rephrased the conclusion sentence. Please see line 432-434:
“Thus, these data, obtained by using different approaches to interfere with mitochondrial dynamics, indicate that circadian variations of neurosteroid production may depend on functional mitochondrial dynamics.”
Reviewer 2 Report
The paper is about the crosstalk between mitochondrial function and circadian cycles. The levels of pregnenolone and translocator protein were investigated. The metodology included "in vitro" and "in vivo" experimentations. The study is interesting, well articulated and well written.
However, the tests carried out on mice raises some doubts about the method used. Since the target of the study was the crosstalk between mithocondrial function and circadian cycles, authors should describe the rationale of the constant darkness for 5 days before the beginning of the experiments on mice. Since light is the main synchronizer of circadian cycles, the choice of 5 days of darkness certainly influenced the results.
In addition, the specifications of the lamp used for 12:12 light-dark cycles should be indicated.
Moreover the method of assessment (and therefore of definition) of "start of subjective daytime" and "start of subjective night-time" (i.e. CT0 and CT12) should be described.
Author Response
However, the tests carried out on mice raises some doubts about the method used. Since the target of the study was the crosstalk between mithocondrial function and circadian cycles, authors should describe the rationale of the constant darkness for 5 days before the beginning of the experiments on mice. Since light is the main synchronizer of circadian cycles, the choice of 5 days of darkness certainly influenced the results.
In addition, the specifications of the lamp used for 12:12 light-dark cycles should be indicated.
Moreover the method of assessment (and therefore of definition) of "start of subjective daytime" and "start of subjective night-time" (i.e. CT0 and CT12) should be described.
Answer: We thank the reviewer for these constructive comments. The aim of our study was to assess whether mitochondrial neurosteroidogenesis follows a circadian rhythm and to investigate the link with circadian variation of mitochondrial morphology. To do so in vivo, it is mandatory that the animals are kept under constant condition by shielding them from external time cues. Thus, these animals display so-called “free running” or circadian rhythms that may persist indefinitely. Circadian rhythms are therefore different from daily rhythms, which are synchronized by external clues such light/dark cycles.
To clarify the methodology behind our in vivo experiments, more details and a new reference were added in the Method part. Please see line 112-124:
“Animal experiments were performed according to standardized experimental designs assessing circadian rhythms in mice [19]. Mice were housed at 24°C on a conventional 12:12 light-dark cycle until the age of 10-12 weeks. Then, they were placed in constant darkness 5 days before the beginning of the experiment with free access to ordinary food (normal chow) and water. Indeed, organisms kept under constant condition (12:12 dark/dark condition) display persisting and intrinsic circadian rhythms (of about 24 hours) that may differ from daily rhythms (rhythms synchronized by external clues, such as light/dark cycles). Under constant dark conditions, the time is expressed in circadian time (CT) units, where CT0 is the start of subjective daytime and CT12 is the start of subjective night-time (corresponding to the onset of activity for nocturnal animals like mice) [20]. Animals were sacrificed every 4 hours over 24 hours and brains were harvested. All experiments were performed in accordance with Swiss animal protection legislation and with approval of the Basel Veterinary Committee for Animal Care: permits 2393 and 23288. The date when the ethical approval was obtained is February 22, 2012 for both permits.”
Round 2
Reviewer 2 Report
The changes made have increased the quality of the paper.
Despite the suggestion, the lamp used for the light exposure of the mice has not been specified, but it is a minor detail.
I have no further comments.